# Astrocytic Nrf2 Mediates the Neuroprotective and Anti-Inflammatory Effects of Nootkatone in an MPTP-Induced Parkinson’s Disease Mouse Model

**DOI:** 10.3390/antiox12111999

**Published:** 2023-11-13

**Authors:** Jung-Eun Park, Yea-Hyun Leem, Jin-Sun Park, Seong-Eun Kim, Hee-Sun Kim

**Affiliations:** 1Department of Molecular Medicine, Inflammation-Cancer Microenvironment Research Center, School of Medicine, Ewha Womans University, Seoul 07804, Republic of Korea; jungeun23@ewha.ac.kr (J.-E.P.); leemyy@ewha.ac.kr (Y.-H.L.); jsp@ewha.ac.kr (J.-S.P.); k2017536@ewhain.net (S.-E.K.); 2Department of Brain & Cognitive Sciences, Ewha Womans University, Seoul 03760, Republic of Korea

**Keywords:** Parkinson’s disease, nootkatone, Nrf2, astrocyte, neuroprotection, anti-inflammation

## Abstract

This study aims to investigate the neuroprotective effects of nootkatone (NKT), a sesquiterpenoid compound isolated from grapefruit, in an MPTP-induced Parkinson’s disease (PD) mouse model. NKT restored MPTP-induced motor impairment and dopaminergic neuronal loss and increased the expression of neurotrophic factors like BDNF, GDNF, and PGC-1α. In addition, NKT inhibited microglial and astrocyte activation and the expression of pro-inflammatory markers like iNOS, TNF-α, and IL-1β and oxidative stress markers like 4-HNE and 8-OHdG. NKT increased the expression of nuclear factor erythroid 2-related factor 2 (Nrf2)-driven antioxidant enzymes like HO-1 and NQO-1 in astrocytes, but not in neurons or microglia in MPTP-treated mice. To investigate whether Nrf2 mediates the anti-inflammatory, antioxidant, or neuroprotective effects of NKT, mice were pretreated with Nrf2-specific inhibitor brusatol (BT) prior to NKT treatment. BT attenuated the NKT-mediated inhibition of 4-HNE and 8-OHdG and the number of Nrf2^+^/HO-1^+^/NQO1^+^ cells co-localized with GFAP^+^ astrocytes in the substantia nigra of MPTP-treated mice. In addition, BT reversed the effects of NKT on dopaminergic neuronal cell death, neurotrophic factors, and pro-/anti-inflammatory cytokines in MPTP-treated mice. Collectively, these data suggest that astrocytic Nrf2 and its downstream antioxidant molecules play pivotal roles in mediating the neuroprotective and anti-inflammatory effects of NKT in an MPTP-induced PD mouse model.

## 1. Introduction

Parkinson’s disease (PD) is the second most prevalent neurodegenerative disease and is characterized by the gradual loss of dopaminergic neurons in the substantia nigra pars compacta (SNpc) [1,2]. A significant loss of dopaminergic neurons can decrease dopamine levels in the striatum, leading to clinical symptoms of motor dysfunction such as resting tremor, muscular rigidity, and bradykinesia [3]. Among the pathogenic components of PD, oxidative stress induces microglial activation, mitochondrial dysfunction, and abnormal α-synuclein aggregation, all of which are pathological characteristics of the disease [4,5]. Indeed, the SNpc of post mortem brains with PD shows enhanced DNA and mitochondrial oxidative damage, lipid peroxidation, and protein oxidation [6,7]. Therefore, reducing oxidative stress has been suggested as a rational strategy for the prevention and treatment of PD [8].

Microglia are the major immune cells in the brain that support neuronal survival by producing neurotrophic factors and clearing apoptotic cells via phagocytosis [9,10]. Astrocytes are the most abundant glial cells in the brain and play a key role in neuronal environment maintenance by providing structural and metabolic support and modulating neurotransmission [11]. However, the activation of microglia or astrocytes leads to a loss of their neuroprotective functions and contributes to neuronal death [12,13]. Activated microglia and astrocytes generate reactive oxygen species (ROS) and release pro-inflammatory cytokines that promote neuroinflammation and neurodegeneration [13,14]. Thus, controlling microglial and astrocyte activation has been proposed as a significant treatment method for neuroinflammatory and neurodegenerative diseases such as PD.

Nuclear factor erythroid 2-related factor 2 (Nrf2) is a redox-sensitive transcription factor that activates several antioxidant genes in response to oxidative stress [15]. Under normal conditions, Nrf2 forms a complex with kelch-like ECH-associated protein 1 (Keap1) and is rapidly destroyed by the ubiquitin–proteasome system. In the presence of oxidative stress and electrophiles, Nrf2 translocates to the nucleus, increasing the antioxidant response element (ARE)-driven transcription of phase II antioxidant enzymes such as hemeoxygenase-1 (HO-1), NAD(P)H:quinone oxidoreductase 1 (NQO1), and manganese superoxide dismutase (MnSOD) [15,16]. Previous studies have shown that Nrf2 has neuroprotective and anti-inflammatory effects on the brain [17,18,19,20,21,22]. Interestingly, ARE-regulated genes are preferentially active in astrocytes, which have better detoxifying and oxidative defenses than neurons [16]. Thus, the activation of Nrf2 in astrocytes has been proposed as a possible therapeutic method not only for acute neuronal injury, but also for chronic neurodegeneration caused by oxidative stress. In the case of spinal cord injury, for example, astrocyte-specific Nrf2 hyperactivation (in GFAP-mediated Keap-1 knockout mice) enhanced locomotor function while decreasing neuroinflammation and demyelination [23]. Furthermore, Nrf2 overexpression in astrocytes protects against amyotrophic lateral sclerosis, Huntington’s disease, multiple sclerosis, and PD [17,24]. Therefore, pharmacological compounds that stimulate Nrf2 signaling, specifically in astrocytes, are regarded as promising treatment options for acute and chronic neurological diseases [23,24].

Nootkatone (NKT; 4,4a,5,6,7,8-hexahydro-6-isopropenyl-4,4a-dimethyl-2(3H) naphthalenone) is a natural sesquiterpenoid compound found in grapefruit, *Alpinia oxyphylla*, and *Cyperus rotundus* plants [25,26]. NKT is widely employed in food additives, cosmetics, and pharmaceuticals because of its numerous pharmacological properties, which include anti-inflammatory, antioxidant, cardioprotective, antiplatelet, and anticancer properties [27,28,29,30,31,32]. Although various researchers have reported on the neuroprotective and anti-inflammatory effects of NKT in neurological disorders such as Alzheimer’s disease (AD), the effect of NKT in a 1-methyl-4-phenyl-1,2,3,6-tetrahydropyridine (MPTP)-induced PD mouse model has not been described until now.

In the present study, we examined the potential neuroprotective and anti-inflammatory effects of NKT in an MPTP-induced PD mouse model. NKT reduced dopaminergic neuronal cell death and neuroinflammation in MPTP-treated mice. NKT increased the Nrf2-driven antioxidant enzyme expression in astrocytes, and the pharmacological inhibition of Nrf2 reversed the neuroprotective and anti-inflammatory effects of NKT. These data suggest that astrocytic Nrf2 plays a key role in mediating the neuroprotective and anti-inflammatory effects of NKT in MPTP-induced PD mice.

## 2. Materials and Methods

### 2.1. Reagents and Antibodies

Nootkatone (NKT) and 1-methyl-4-phenylpyridinium ion (MPP^+^) were purchased from Sigma-Aldrich (St. Louis, MO, USA). Brusatol (BT) and MPTP were purchased from MedChemExpress (Princeton, NJ, USA) and Tokyo Chemical Industry Co., Ltd. (Tokyo, Japan), respectively. The following primary antibodies were used in this study: anti-tyrosine hydroxylase (TH), anti-p-cAMP response element-binding protein (CREB), and anti-NQO1 from Cell Signaling Technology, Inc. (Danvers, MA, USA); anti-B-cell lymphoma 2 (Bcl-2) and anti-glial cell-derived neurotrophic factor (GDNF) from Abcam (Cambridge, UK); anti-ionized calcium-binding adapter molecule1 (Iba-1) from Wako (Osaka, Japan); anti-inducible nitric oxide synthase (iNOS) from BD Biosciences (San Jose, CA, USA); anti-tumor necrosis factor (TNF-α), anti-interleukin (IL)-6, anti-Toll-like receptor 2 (TLR2), anti-TLR4, anti-IL-10, anti-Nrf2, and anti-transforming growth factor-beta (TGF-β) from Santa Cruz Biotechnology (Santa Cruz, CA, USA); anti-peroxisome proliferator-activated receptor-gamma coactivator-1 alpha (PGC-1α) and anti-neuronal nuclear (NeuN) from Merck-Millipore (Billerica, MA, USA); anti-HO-1 and anti-MnSOD from Enzo Life Sciences (Farmingdale, NY, USA); anti-4-hydroxy-2*E*-nonenal (4-HNE) from Alpha diagnostic Intl Inc. (San Antonio, TX, USA); anti-8-hydroxy-2′-deoxyguanosine (8-OHdG) from Antibodies-online (Aachen, German); anti-γ-glutamyl cysteine ligase-modulator subunit (GCLM) from GeneTex (Irvine, CA, USA); anti-cyclooxygenase-2 (COX-2), anti-IL-1β, and anti-γ-glutamyl cysteine ligase-catalytic subunit (GCLC) from MyBioSource (San Diego, CA, USA); and anti-glial fibrillary acidic protein (GFAP), anti-brain-derived neurotrophic factor (BDNF), and anti-β-actin from Sigma-Aldrich. The detailed catalogue numbers and dilution folds of each antibody are summarized in Appendix A.

### 2.2. Primary Astrocyte Culture

Primary astrocyte cultures were produced from mixed glial cultures using a modified version of a previously described method [33]. Briefly, after dissecting cortices from 1-day-old rats, the cells were dissociated and resuspended in media. Cell suspensions were plated on T75 flasks coated with poly-D-lysine (1 g/mL) and incubated at 37 °C with 5% CO_2_. The culture flasks were centrifuged at 280 rpm for 16 h four days later to eliminate the microglia and oligodendrocytes. The leftover astrocytes were trypsinized, seeded onto a culture plate, and cultured for 3–6 days. The quality of the astrocyte cultures was greater than 95%, as determined by Western blotting and immunocytochemical analyses with an anti-GFAP antibody.

### 2.3. Animals

Adult male C57BL/6 mice (8 weeks old) were procured from Orient Bio Inc., a Charles River Laboratories affiliate in Seongnam, Korea. The mice were housed in an environment with a temperature of 21 °C under a 12 h light/dark cycle with ad libitum access to water and rodent chow. All experiments were carried out in accordance with the National Institutes for Health and Ewha Womans University guidelines for the Care and Use of Laboratory Animals, and the study was approved by the Ewha Womans University Medical School’s Institutional Animal Care and Use Committee (EWHA MEDIACUC 21-001-2).

### 2.4. Drug Administration

Mice were randomly divided into control, MPTP, MPTP + NKT, MPTP + NKT + BT, BT, and NKT groups (each group n = 10–12). The MPTP-induced PD mouse model was established as described in our previous reports [34]. Briefly, BT (1 mg/kg, i.p.) was injected 6 h before NKT (2 or 5 mg/kg, i.p.) for three consecutive days. BT and NKT were dissolved in 1% DMSO in a saline solution. One day after the final NKT treatment, MPTP (20 mg/kg, i.p.) was administered four times at 2 h intervals, and the mice were sacrificed 7 days after MPTP administration. The MPTP model is commonly used to study the pathophysiology of PD because of its convenience and similarities to PD pathology (loss of dopaminergic neurons and neuroinflammation), despite its limitations, which include the inability to represent age-related changes, the absence of Lewy body development, and the lack of progressive dopaminergic neuronal death.

### 2.5. Assessment of Motor Function

Rotarod and pole tests were performed to assess motor function in mice, as previously described [35]. Two and six days after receiving MPTP injections, mice were submitted to behavioral tests (rotarod test, two days; pole test, six days). On the test day, the mice were put on the resting drum of a rotarod device (Harvard Apparatus, Holliston, MA, USA) for at least 1 min. The rotation speed was increased from 4 to 40 rpm over 300 s. The mice were put through three trials with 15 min intervals between trials. In each trial, the retention time of the rods was recorded. For the pole test, all the mice were trained to successfully descend from the pole’s peak to its base. The time it took each mouse to descend the pole was recorded. Each mouse was given three trials, with the average being recorded.

### 2.6. Immunohistochemistry and Immunofluorescence Staining

Mice were euthanized with sodium pentobarbital and perfused with saline to remove blood from the brain. After that, the brains were separated and cryoprotected in a 30% sucrose solution at 4 °C, and 40 μm thick brain sections were prepared using a cryotome (CM1860; Leica, Mannheim, Germany). Then, immunohistochemistry (IHC) and immunofluorescence (IF) staining were carried out, as previously described [34,35]. For IHC, the sections were treated overnight with primary antibodies, then for 1 h at 25 °C with biotinylated secondary antibodies, followed by 1.5 h with avidin-biotin-HRP complex reagent solution (Vector Laboratories, Burlingame, CA, USA). The peroxidase reaction was then carried out using a diaminobenzidine tetrahydrochloride solution (Vector Laboratories). For IF staining, the sections were treated with 4% BSA and incubated with primary antibodies at 4 °C overnight, followed by fluorochrome-conjugated secondary antibodies (Alexa Fluor 488 and 594). The slides were then covered with VECTASHIELD antifade mounting medium (Vector Laboratories). To collect digital images for IHC and IF staining, a Leica DM750 microscope was employed, and the ImageJ software (version 1.37) was used for quantification (NIH, Bethesda, MD, USA).

### 2.7. Reverse-Transcription Polymerase Chain Reaction (RT-PCR)

Total RNA was isolated from primary astrocytes or mouse brain tissue using TRIzol reagent (Invitrogen, Carlsbad, CA, USA). For RT-PCR, total RNA (1 μg) was reverse transcribed in a reaction mixture containing 500 ng random primer, 0.5 mM dNTP, 3 mM MgCl_2_, 1X RT buffer, 10 U reverse transcriptase, and 1 U RNase inhibitor. The cDNA was generated and utilized as a template for PCR with primers and Go Taq polymerase (Promega). RT-PCR was performed on a T100 Thermal cycler (Bio-Rad Laboratories, Hercules, CA, USA). The ABI PRISM 7000 Sequence Detection System (Applied Biosystems, Foster City, CA, USA) and Sensi FASTTM SYBR Hi-ROX Mix (Bioline, London, UK) were used for quantitative RT-PCR. To normalize the expression levels of the target genes towards GADPH, the following formula was utilized: 2^(Ct, test gene–Ct, GAPDH)^. Table 1 shows the primer sequences used for PCR.

### 2.8. Western Blot Analysis

Whole-cell protein lysates from brain tissue homogenates or primary astrocytes were prepared in a lysis buffer (Roche, Basel, Switzerland). Following that, the samples were vortexed vigorously and incubated at 4 °C for 30 min. After centrifuging the samples at 20,000× *g* for 30 min, the supernatant was recovered. SDS-PAGE was used to separate protein samples (30–50 μg) and transfer them to a nitrocellulose membrane. The blots were blocked in TBST with 5% skim milk for 1 h and incubated with primary antibodies according to the manufacturer’s instructions. After thoroughly washing the membranes with TBST, they were incubated with horseradish peroxidase-conjugated secondary antibodies (Bio-Rad, Hercules, CA, USA; 1:3000 dilution in TBST). Following that, the blots were developed using an enhanced chemiluminescence detection kit (Thermo Fisher Scientific). Using ImageJ software, the density of certain target bands was normalized against β-actin for measurement.

### 2.9. Detection of Intracellular ROS Levels

Primary astrocytes (2 × 10^5^ cells per well on a 24-well plate) were pretreated for 1 h with NKT and then activated for 16 h with MPP^+^ (1 mM). To determine the intracellular ROS levels, the cells were loaded with CellROX^TM^ green reagent (5 μM; Thermo Fisher, Waltham, MA, USA) and incubated for 30 min. Digital images were captured using a Leica DM750 microscope, and quantification was performed using ImageJ software.

### 2.10. Measurement of Glutathione Levels

Glutathione (GSH) levels were determined using the DTNB (5,5′-dithiobis-[2-nitrobenzoic acid]) technique using a QuantiChrom^TM^ Glutathione Assay kit (DIGT-250, BioAssay Systems, Hayward, CA, USA). Briefly, brain tissue or cell lysates were homogenized in a cold buffer containing 1 mM EDTA (pH 6.0) and 50 mM MES, and the supernatants were deproteinized and mixed with DTNB. The optical density was measured at 412 nm after 25 min of incubation at room temperature. All measurements were normalized to the amount of protein in each sample.

### 2.11. Transient Transfection and Luciferase Assay

Primary astrocytes (2 × 10^5^ cells per well on a 12-well plate) were transfected with a reporter plasmid (ARE-luc, HO-1 E1-luc, NQO1-luc) using the Metafectene transfection reagent (Biontex, Munich, Germany). After 36 h of transfection, the cells were pretreated with BT for 1 h and then treated for 6 h with NKT. The cells were collected and the luciferase assay was carried out as previously described [36]. Small interfering RNA (siRNA) targeting rat Nrf2 mRNA and scrambled control siRNA were obtained from Santa Cruz Biotechnology. Nrf2 siRNA (sc-156128) was transfected into primary astrocytes using the Metafectene transfection reagent.

### 2.12. Electrophoretic Mobility Shift Assay (EMSA)

Nuclear extracts were obtained from primary astrocytes following 3 h of NKT treatment, as previously described [37]. T4 polynucleotide kinase (New England Biolabs, Beverly, MA, USA) and [γ-^32^P] ATP were used to end-label the double-stranded DNA oligonucleotides containing ARE consensus sequences (Santa Cruz Biotechnology). Nuclear proteins (10 μg) were incubated with a ^32^P-labeled probe on ice for 30 min before being resolved on a 5% acrylamide gel.

### 2.13. Statistical Analysis

One-way analysis of variance (ANOVA) was used to assess differences between experimental groups, and the least significant difference (LSD) test was used for post hoc comparisons. All statistical analyses were performed using the SPSS for Windows (version 18.0 (SPSS Inc., Armonk, NY, USA). The sample size was not predetermined for the in vivo experiments. All values are reported as the mean ± standard error of the mean (SEM). Statistical significance was set at *p* value < 0.05.

## 3. Results

### 3.1. NKT Inhibited Dopaminergic Neuronal Cell Death and Restored the Expression of Tyrosine Hydroxylase (TH) and Neurotrophic Factors in MPTP-Treated Mice

To investigate whether NKT has neuroprotective effects, mice were given NKT for three days before MPTP injection and euthanized seven days later for analysis. Figure 1A depicts the experimental process. We chose three days of NKT treatment before MPTP administration because prior research showed that three consecutive days of pharmacological treatment provided more effective and consistent effects than a single bout of treatment. In terms of dosing, we discovered that 10–50 mg/kg of NKT induced significant mortality in MPTP-treated mice. As a result, in this study, we utilized 2–5 mg/kg as an optimum dosage. As shown in Figure 1B, NKT administration decreased the MPTP-induced elongation of the descending duration in the pole test and improved the MPTP-induced decline in rod retention time in the rotarod test. These results indicate that NKT improved MPTP-induced motor impairment and akinesia. NKT prevented dopaminergic neuronal cell death in the substantia nigra (SN) and restored dopaminergic neuronal fibers in the striatum, according to immunohistochemistry using a TH antibody (Figure 1C). Western blot analysis validated these findings by showing that the MPTP injection decreased TH levels, which were recovered with NKT (Figure 1D). Moreover, MPTP injection decreased the levels of neurotrophic factors such as p-CREB, PGC-1α, BDNF, GDNF, and Bcl-2 in the SN, which were all restored by NKT treatment (Figure 1D).

### 3.2. NKT Exerted Anti-Inflammatory and Antioxidant Effects by Inhibiting the Activation of Astrocytes and Microglia in MPTP-Treated Mice

The excessive activation of glial cells accelerates the release of neurotoxic factors, leading to neuronal cell death [13]. Oxidative stress and neuroinflammatory responses can damage the dopaminergic neurons [5]. In this study, we found that NKT suppressed MPTP-induced astrocytes and microglial activation in the striatum and SN (Figure 2). In addition, NKT suppressed the expression of pro-inflammatory markers such as iNOS, COX-2, IL-1β, TNF-α, IL-6, Iba-1, GFAP, and TLR-2/4, induced by MPTP treatment, while it increased the anti-inflammatory cytokines IL-10 and TGF-β (Figure 3A–D). NKT also suppressed the MPTP-induced production of 4-HNE, a lipid peroxidation product caused by oxidative stress, and restored the protein levels of Nrf2 and its downstream antioxidant enzymes, such as HO-1, NQO1, and MnSOD, which were reduced by MPTP treatment (Figure 3E,F). Moreover, NKT restored the expression of GCLC and GCLM and its downstream antioxidant GSH (Figure 3E–G).

### 3.3. NKT Increased Nrf2-Driven Antioxidant Enzymes in the Astrocytes of MPTP-Treated Mice and Rat Primary Astrocytes

Double IF staining was performed to identify the source cell types for Nrf2, HO-1, and NQO1 expression in NKT/MPT-treated mice. MPTP treatment significantly reduced the expression of Nrf2, HO-1, and NQO1 and the number of Nrf2^+^, HO-1^+^, and NQO1^+^ cells that co-localized with astrocytes in the SN region, whereas NKT restored the expression of Nrf2, HO-1, and NQO1 in astrocytes (Figure 4A–C). However, NKT treatment had no significant effect on the number of Nrf2^+^ cells co-localized with microglia and only slightly increased the number of Nrf2^+^ cells co-localized with neurons (Appendix A). In accordance with this, NKT treatment had no significant effect on the number of HO-1^+^ or NQO1^+^ cells co-localized with microglia or neurons, as shown by Iba-1 or NeuN staining (Appendix A). Next, we examined the antioxidant effects of NKT on primary rat astrocytes. IF staining revealed that NKT inhibited ROS production in MPP^+^-treated astrocytes (Figure 5A). To verify the effects of NKT on Nrf2 signaling, rat primary astrocytes were treated with NKT, and the expressions of antioxidant enzymes and Nrf2 activity were measured. Western blot and RT-PCR analyses showed that NKT induced the expression of HO-1, NQO1, and MnSOD at the mRNA and protein levels (Figure 5B,C). Moreover, NKT increased the intracellular GSH levels and the expression of GCLC and GCLM, which are the rate-limiting enzymes in GSH synthesis (Figure 5D–F). When we examined the effect of NKT on upstream Nrf2 signaling, we found that NKT increased the nuclear translocation, DNA binding, and transcriptional activity of Nrf2 in primary astrocytes (Figure 5G–I).

### 3.4. Pharmacological Inhibition or Knockdown of Nrf2 Showed That the Nrf2/ARE Signaling Pathway Mediates Antioxidant Enzyme Expression in NKT-Treated Astrocytes

To verify that Nrf2 modulates NKT-induced antioxidant enzyme expression, Nrf2 knockdown experiments using siRNA were performed on primary astrocytes. We confirmed that Nrf2 siRNA, but not control siRNA, significantly blocked Nrf2 expression (Figure 6A). Western blot analysis showed that Nrf2 siRNA abolished the NKT-mediated upregulation of HO-1, NQO1, MnSOD, GCLC, and GCLM (Figure 6B). To further investigate the involvement of Nrf2 signaling in the effects of NKT, astrocytes were treated with brusatol (BT, an Nrf2-specific inhibitor) [38] before NKT treatment. Consistent with the siRNA experiments, BT suppressed the NKT-induced expression of HO-1, NQO1, MnSOD, GCLC/GCLM, and GSH (Figure 6C,D). In addition, BT suppressed the NKT-induced expression of antioxidant enzymes at the mRNA level (Figure 6E). Furthermore, BT reversed the effects of NKT on the transcriptional activity of ARE-luc, HO-1 E1-luc, and NQO1-luc (Figure 6F). Collectively, these data suggest that Nrf2 is a key regulator of antioxidant enzyme gene expression in NKT-treated astrocytes. Although the astrocyte culture system has some advantages, such as relative simplicity, cell specificity, and feasible experimental control, it also has some limitations, including the inability of primary astrocytes to realize cell-to-cell interactions in the brain (neuron–astrocyte–microglia) and region-specific responsiveness. As a result, in the next experiments, we verified the findings from primary astrocytes in MPTP-treated mice.

### 3.5. The Nrf2 Inhibitor Brusatol Reverses the Effects of NKT on Oxidative Stress and Astroglial Antioxidant Enzyme Expression in MPTP-Treated Mice

To investigate whether Nrf2 also controls oxidative stress and antioxidant enzyme expression in vivo, mice were administered BT 6 h prior to NKT injection. Western blot analysis showed that BT reversed the NKT-mediated inhibition of 4-HNE and the upregulation of Nrf2, HO-1, NQO1, GCLC, GCLM, and GSH levels in the SN (Figure 7A,B). IF staining showed that NKT inhibited the production of 8-OHdG, a product of oxidative damage from 2-deoxyguanosine, which is widely utilized as an oxidative stress marker, in astrocytes of the SN (Figure 7C–E). BT reversed the inhibitory effect of NKT on 8-OHdG levels, suggesting that Nrf2 mediates the antioxidant effect of NKT. Next, we examined the effects of BT on astrocytic Nrf2, HO-1, and NQO1 expression in MPTP/NKT-treated mice. Double IF staining showed that BT reversed NKT-induced Nrf2 expression and reduced the number of Nrf2^+^/GFAP^+^ cells in the SN region (Figure 7F). Similarly, NKT-induced HO-1 and NQO1 expression, and the number of HO-1^+^/GFAP^+^ and NQO1^+^/GFAP^+^ cells, were reversed by BT treatment (Figure 8A,B). These results suggest that Nrf2 mediates NKT-induced HO-1 and NQO1 expression in astrocytes of MPTP mice.

### 3.6. The Nrf2 Inhibitor Brusatol Reverses the Neuroprotective and Anti-Inflammatory Effects of NKT in MPTP-Treated Mice

To see if the Nrf2 signaling pathway is responsible for the neuroprotective and anti-inflammatory effects of NKT in MPTP mice, mice were given BT prior to NKT injection, and behavioral, immunohistochemical, and biochemical investigations were carried out. The rotarod and pole tests demonstrated that BT reversed the NKT-mediated improvement in motor function in MPTP mice (Figure 9A). In addition, BT reversed the neuroprotective effects of NKT on dopaminergic neurons in the SN and striatum, as shown by the IHC analysis using the TH antibody (Figure 9B). Furthermore, BT reversed the NKT-mediated inactivation of astrocytes and microglia in MPTP mice (Figure 9C,D). Western blot analysis using brain tissue from the SN region showed that BT reversed the NKT-mediated recovery of protein levels of TH, as well as neurotrophic factors including BDNF, GDNF, and Bcl-2 (Figure 10A). BT also reversed the NKT-mediated anti-inflammatory effects by increasing pro-inflammatory factors, such as iNOS, TNF-α, IL-6, and IL-1β, and decreasing anti-inflammatory cytokines, such as IL-10 and TGF-β in the SN of MPTP-injected mice (Figure 10B). These findings imply that Nrf2 plays a pivotal role in mediating the neuroprotective and anti-inflammatory effects of NKT in an MPTP-induced PD mouse model.

## 4. Discussion

The current study demonstrated the neuroprotective and anti-inflammatory effects of NKT and the underlying molecular mechanisms in an MPTP-induced PD mouse model. NKT treatment ameliorated motor impairment and inhibited dopaminergic neuronal death caused by the neurotoxic effects of MPTP. NKT also suppressed the expression of inflammatory factors by inhibiting microglial and astrocyte activation. In addition, NKT increased GSH synthesis and the expression of Nrf2-driven antioxidant enzymes including HO-1, NQO1, and MnSOD. NKT-mediated activation of antioxidant enzymes was observed in astrocytes, but not in neurons or microglia in the SN of MPTP-treated mice. Using an Nrf2-specific inhibitor, we demonstrated that astrocytic Nrf2 mediates the neuroprotective and anti-inflammatory effects of NKT in mice with MPTP-induced PD. Although various pharmacological inducers of Nrf2 have shown neuroprotective effects in PD and other neurodegenerative diseases, the cell types targeted by these drugs have not been clearly identified. Our findings are unique because we discovered that the mechanism of action of NKT is astrocyte-specific.

A recent study by our group reported that NKT exerts anti-inflammatory effects on LPS-induced neuroinflammation [36]. NKT reduced microglial activation, lipid peroxidation, and the expression of pro-inflammatory markers in the brains of LPS-injected mice. We found that NQO1 plays an important role in mediating the anti-inflammatory and antioxidant effects of NKT through modulating AMPK and its downstream signaling pathways. Moreover, neuroprotective effects of NKT have been reported in AD and anxiety/depression models [29,30,31]. NKT improved cognitive impairment by attenuating neuroinflammation in an LPS-induced mouse model of AD [29]. In addition, NKT ameliorated Aβ-induced memory impairment and improved hippocampal neuronal damage via antioxidative and anticholinesterase activities [30]. NKT also improved anxiety- and depression-like behaviors by alleviating oxidative stress through the Nrf2/ARE pathway in a D-galactosamine model of liver injury [31]. Despite these findings, the therapeutic effects of NKT in an MPTP-induced PD mouse model and its detailed molecular mechanisms have not yet been demonstrated. In this study, we demonstrated for the first time that NKT possesses neuroprotective, anti-inflammatory, and antioxidant effects in an MPTP-induced PD mouse model and that astrocytic Nrf2 plays a key role in mediating these effects. A recent study has shown that NKT penetrates the blood–brain barrier (BBB) and enters the blood, cerebrospinal fluid, and brain tissue [32]. Considering its BBB permeability and minimal side effects, NKT may be a promising agent for the treatment of PD and other neurodegenerative diseases.

Nrf2 activation in astrocytes has been proposed as a potential therapeutic approach for neurodegenerative diseases because it increases antioxidant defense, mitochondrial biogenesis, and autophagy capacity while inhibiting neuroinflammation and oxidative stress [18,39]. In the present study, we found that NKT, a pharmacological Nrf2 activator, upregulated target proteins, such as HO-1, NQO1, MnSOD, GCLC, GCLM, and GSH, in astrocytes in vivo and in vitro. HO-1 protects cells from oxidative damage via the breakdown of the pro-oxidant heme group to the radical-scavenging bile pigments, biliverdin (BV) and bilirubin (BR), and carbon monoxide (CO) [40]. BV is converted into BR by biliverdin reductase. BR has antioxidant, anti-inflammatory, and neuroprotective properties, and has recently been suggested as a promising therapy for PD [41]. CO promotes the resolution of inflammation by upregulating the synthesis of pro-resolving mediators and by downregulating inflammasome activation [40]. NQO1 inhibits the production of free radicals by catalyzing the two-electron reduction of quinone to the redox-stable hydroxyquinone by interacting with NADH, which boosts intracellular NAD^+^ levels. Antioxidant, anti-inflammatory, and cytoprotective effects of NQO1 have been reported under neurological conditions [36,42]. SOD is a key defense mechanism that converts superoxide to hydrogen peroxide in order to counteract oxygen toxicity [43]. GCLC and GCLM are rate-limiting enzymes that regulate the synthesis of GSH, which reacts with radicals in the extracellular space and is the first line of defense against ROS [42]. GSH is hydrolyzed by extracellular enzymes to cysteine and glycine, which are then taken up by neurons to form GSH, which plays a neuroprotective role [44,45].

Based on these findings, we propose a plausible mechanism for NKT’s actions in MPTP-treated mice (Figure 11). MPTP administration increases astrocyte activation and decreases Nrf2/ARE signaling and antioxidant enzyme expression. The inflammatory mediators secreted from reactive astrocytes promote the activation of microglia and the production of ROS and pro-inflammatory factors. In astrocytes, MPTP is converted into the toxic metabolite MPP^+^, which is taken up by dopaminergic neurons via dopamine transporters. MPP^+^ then increases ROS production through mitochondrial complex I inhibition, while decreasing neurotrophic factor expression. Furthermore, the release of neurotoxic factors by activated microglia and astrocytes promotes dopaminergic neuronal cell death, leading to locomotor dysfunction. NKT treatment enhances astroglial Nrf2 signaling and the expression of HO-1, NQO1, MnSOD, and GSH. The antioxidant molecules may operate together to reduce ROS production in astrocytes and neighboring neuronal and microglial cells, resulting in the inhibition of neuroinflammation, dopaminergic neuronal death, and subsequent locomotor dysfunction. The Nrf2-specific inhibitor, brusatol, dramatically inhibited all the aforementioned actions of NKT, indicating that Nrf2 functions as a master regulator of the neuroprotective, anti-inflammatory, and antioxidant effects of NKT.

## 5. Conclusions

To our knowledge, this is the first investigation into the neuroprotective, anti-inflammatory, and antioxidant effects of the pharmacological Nrf2 activator NKT in an MPTP-induced PD model, as well as on the crucial involvement of astrocytic Nrf2 in the NKT action mechanism. Given its BBB permeability and lack of adverse effects, NKT could be a good candidate molecule for the treatment of PD and other neurodegenerative disorders related to oxidative stress.

## Figures and Tables

**Figure 1 antioxidants-12-01999-f001:**
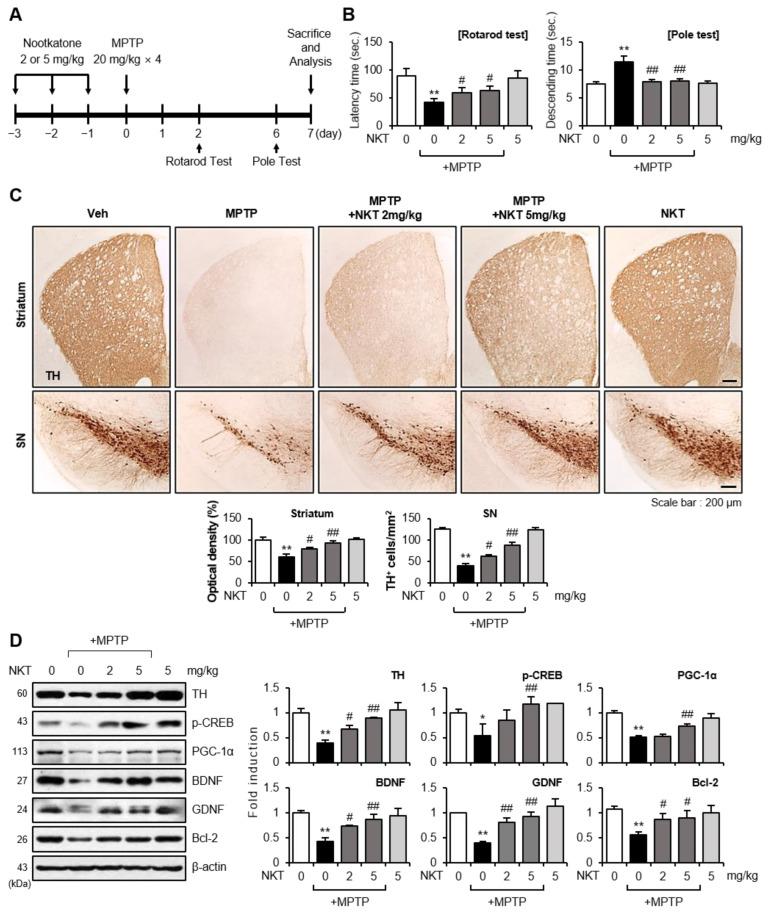
Effects of NKT on locomotor activity, dopaminergic neuronal cell death, and neurotrophic factor expression in the brains of MPTP-treated mice. (**A**) The experimental process is depicted schematically. Mice were given NKT (2 or 5 mg/kg, i.p.) for three days prior to MPTP injection. After 7 days of MPTP treatment, the mice were killed, and histological and biochemical analyses were performed. (**B**) Two and six days after MPTP injection, rotarod and pole tests were performed (n = 10–12 per group). (**C**) Immunohistochemical (IHC) staining for TH in the striatum and substantia nigra (SN) (n = 4–5 per group, 3 sections per brain). The number of TH^+^ cells in the SN and the optical density of TH^+^ fibers in the striatum were measured quantitatively (bottom panel). (**D**) Western blot analysis to evaluate neurotrophic factor protein expression in the SN of each group using TH, p-CREB, PGC-1α, BDNF, GDNF, and Bcl-2 antibodies (each group n = 6–7). Representative blots are shown in the left panel, and quantification data are shown in the right panel. Data are presented as the mean ± SEM. * *p* < 0.05 vs. control group; ** *p* < 0.01 vs. control group; ^#^
*p* < 0.05 vs. MPTP-treated group; ^##^
*p* < 0.01 vs. MPTP-treated group.

**Figure 2 antioxidants-12-01999-f002:**
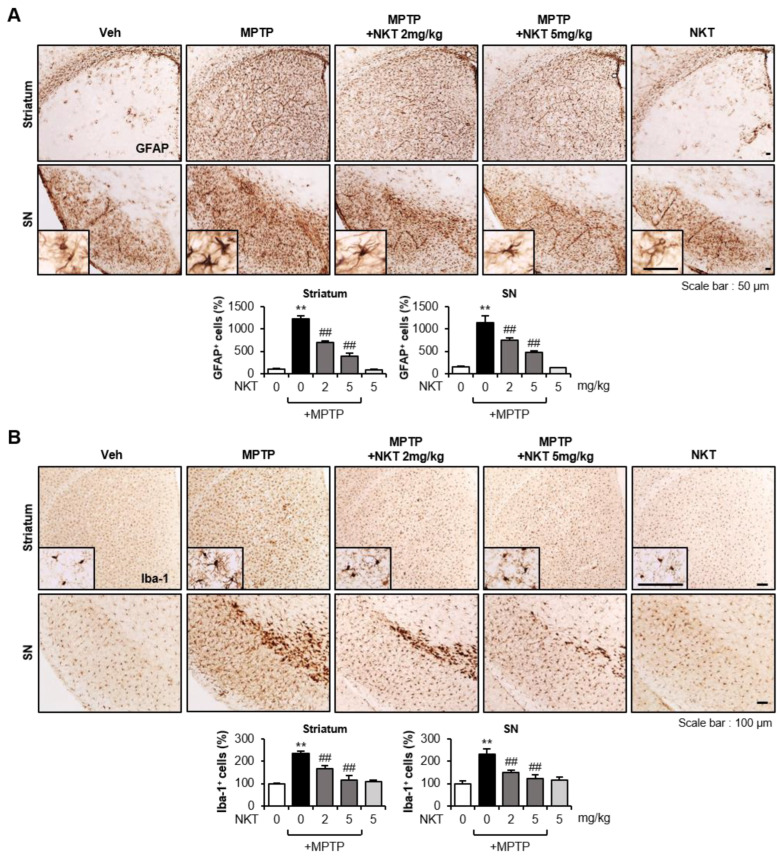
Effects of NKT on astrocyte and microglial activation in the brains of MPTP-treated mice. (**A**) IHC staining for GFAP (astrocyte marker) in the striatum and SN (n = 4–5 per group, 3 sections/brain). Quantitative analysis was performed by measuring the number of GFAP^+^ cells (bottom panel). (**B**) IHC staining for Iba-1 (microglial marker) in the striatum and SN (n = 4–5 per group, 3 sections/brain). The number of Iba-1^+^ cells was counted to undertake quantitative analysis (bottom panel). The data are presented as the mean ± SEM. ** *p* < 0.01 vs. control group; ^##^
*p* < 0.01 vs. MPTP-treated group.

**Figure 3 antioxidants-12-01999-f003:**
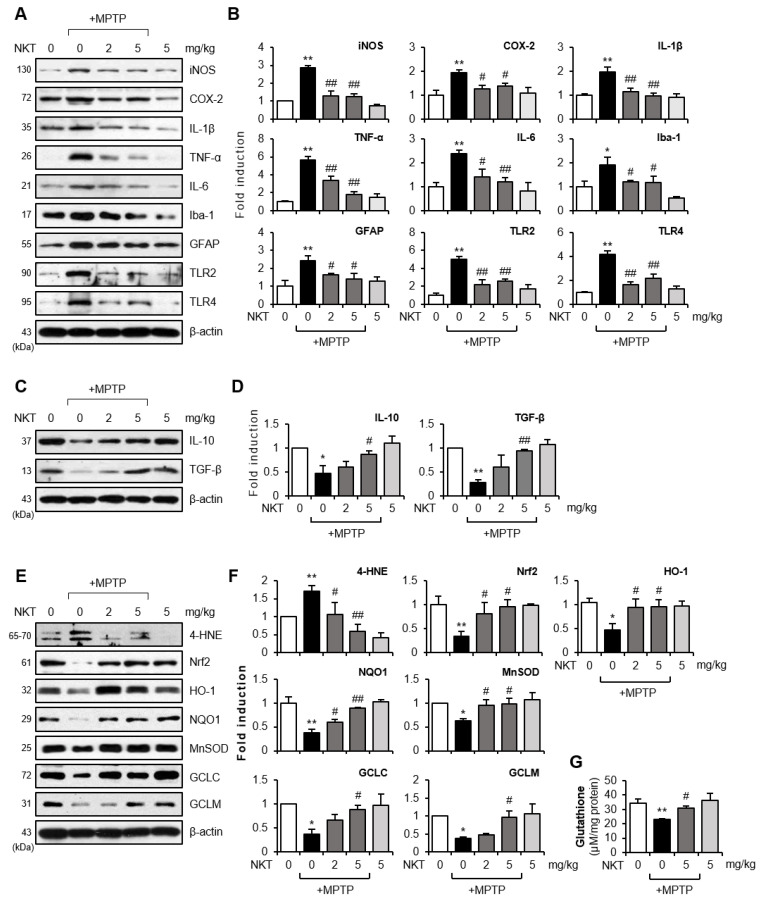
Effects of NKT on the expressions of pro-inflammatory, anti-inflammatory, and antioxidant molecules in the brains of MPTP-treated mice. (**A**) Western blot data showing the effects of NKT on the protein expressions of iNOS, COX-2, pro-inflammatory cytokines, Iba-1, GFAP, TLR2/4 in the SN of MPTP mice (each group n = 6–7). (**B**) Quantification of Western blot data. (**C**) Effects of NKT on anti-inflammatory cytokines such as TGF-β and IL-10 in the SN of MPTP mice (each group n = 6–7). (**D**) Quantification of Western blot data. (**E**) Western blot data showing the effects of NKT on 4-HNE, Nrf2, HO-1, NQO1, MnSOD, GCLC, and GCLM in the SN of MPTP mice (each group n = 6–7). (**F**) Quantification of Western blot data. (**G**) Effect of NKT on intracellular GSH level in the SN of MPTP mice. The data are presented as the mean ± SEM. * *p* < 0.05 vs. control group; ** *p* < 0.01 vs. control group; ^#^
*p* < 0.05 vs. MPTP-treated group; ^##^
*p* < 0.01 vs. MPTP-treated group.

**Figure 4 antioxidants-12-01999-f004:**
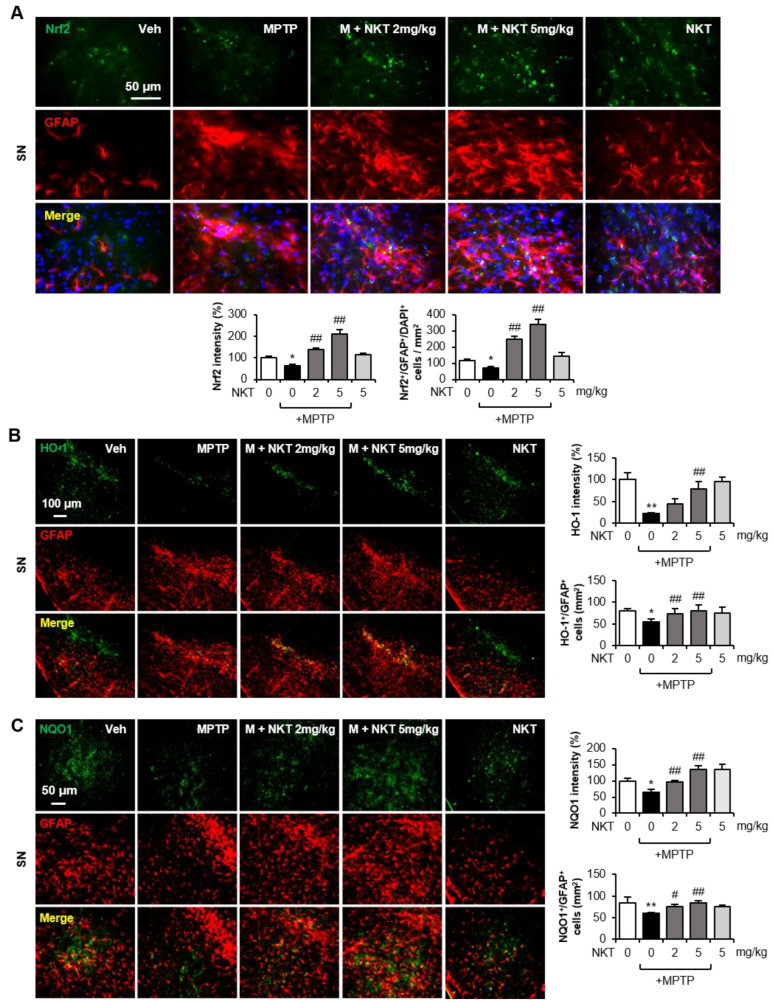
NKT increased astroglial Nrf2, HO-1, and NQO1 expression in the brains of MPTP-treated mice. (**A**) Immunofluorescence (IF) staining results showing Nrf2 and GFAP expression in the SN of MPTP mice (n = 6–7, 3 sections/brain). The upper panel has representative images, and the quantification of Nrf2 expression and Nrf2^+^/GFAP^+^/DAPI^+^ cells is shown in the bottom panel. (**B**,**C**) IF staining results showing HO-1/GFAP (**B**) and NQO1/GFAP (**C**) expression in the SN of MPTP mice (n = 6–7, 3 sections/brain). Representative images are shown in the left panel, and the quantification of HO-1, NQO1 expression, and HO-1^+^/GFAP^+^, NQO1^+^/GFAP^+^ cells is shown in the right panel. The data are presented as the mean ± SEM. * *p* < 0.05 vs. control group; ** *p* < 0.01 vs. control group; ^#^
*p* < 0.05 vs. MPTP-treated group; ^##^
*p* < 0.01 vs. MPTP-treated group.

**Figure 5 antioxidants-12-01999-f005:**
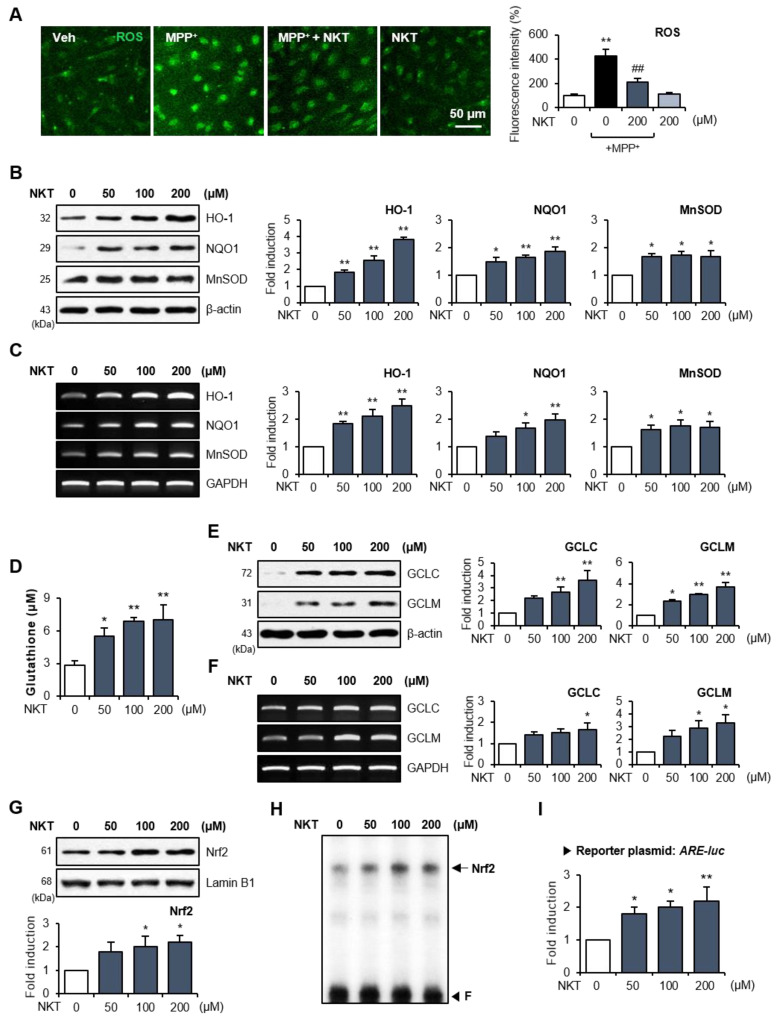
NKT inhibited ROS production and upregulated the antioxidant enzyme expression via Nrf2/ARE signaling pathway in rat primary astrocytes. (**A**) Rat primary astrocytes were treated with NKT prior to MPP^+^ stimulation for 16 h, and intracellular ROS level was evaluated with CellROX staining (n = 3). Representative fluorescence images are shown in the left panel, and the quantification of fluorescence intensity is shown in the right panel. (**B**) Astrocyte cells were incubated with NKT for 6 h to determine the effects of NKT on the protein expression of HO-1, NQO1, and MnSOD (n = 3). (**C**) RT-PCR results demonstrating the influence of NKT on mRNA expression of antioxidant enzymes (n = 3). (**D**) Effect of NKT on intracellular GSH level in astrocytes (n = 3). (**E**) Effect of NKT on GCLC and GCLM protein expression (n = 3). Quantification data of Western blot analysis are shown in right panel. (**F**) RT-PCR results demonstrating the influence of NKT on mRNA expression of GCLC and GCLM (n = 3). (**G**) Western blot data showing the effect of NKT on nuclear translocation of Nrf2. (**H**) EMSA data showing the effect of NKT on Nrf2 DNA binding activity. ‘F’ indicates free probe. (**I**) Effect of NKT on ARE-luc reporter gene activity. The data are presented as the mean ± SEM. * *p* < 0.05 vs. control cells; ** *p* < 0.01 vs. control cells; ^##^
*p* < 0.01 vs. MPP^+^-treated cells.

**Figure 6 antioxidants-12-01999-f006:**
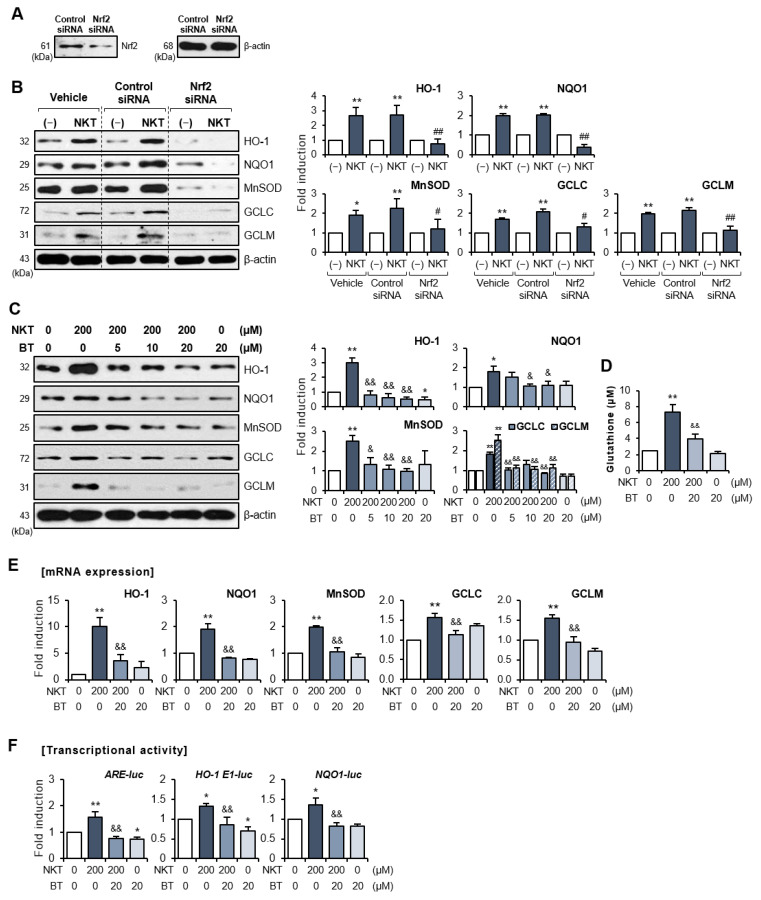
Nrf2 siRNA or Nrf2-specific inhibitor, brusatol, reversed NKT-mediated upregulation of antioxidant molecules in rat primary astrocytes. (**A**,**B**) Cells were transfected with Nrf2 siRNA or control siRNA and then treated with NKT for 6 h to detect Nrf2 and antioxidant enzyme levels (n = 3). Western blot analysis was performed to confirm Nrf2 knockdown (**A**) and the expression of antioxidant enzymes (**B**). The left panel shows representative blots, and the right panel shows quantification data. * *p* < 0.05 vs. control cells; ** *p* < 0.01 vs. control cells; ^#^
*p* < 0.05 vs. control siRNA-transfected cells in the presence of NKT; ^##^
*p* < 0.01 vs. control siRNA-transfected cells in the presence of NKT. (**C**,**D**) Cells were incubated with BT for 1 h, followed by NKT for 6 h to detect antioxidant enzymes (**C**) and GSH levels (**D**). The right panel depicts the quantification of Western blot data. (**E**) Quantitative RT-PCR data showing the effect of NKT on mRNA expressions of antioxidant enzymes and GCLC/GCLM. (**F**) Effect of NKT on transcriptional activities of ARE-luc, HO-1 E1-luc, or NQO1-luc (n = 3). The data are presented as the mean ± SEM. * *p* < 0.05 vs. control cells; ** *p* < 0.01 vs. control cells; ^&^
*p* < 0.05 vs. NKT-treated cells; ^&&^
*p* < 0.01 vs. NKT-treated cells.

**Figure 7 antioxidants-12-01999-f007:**
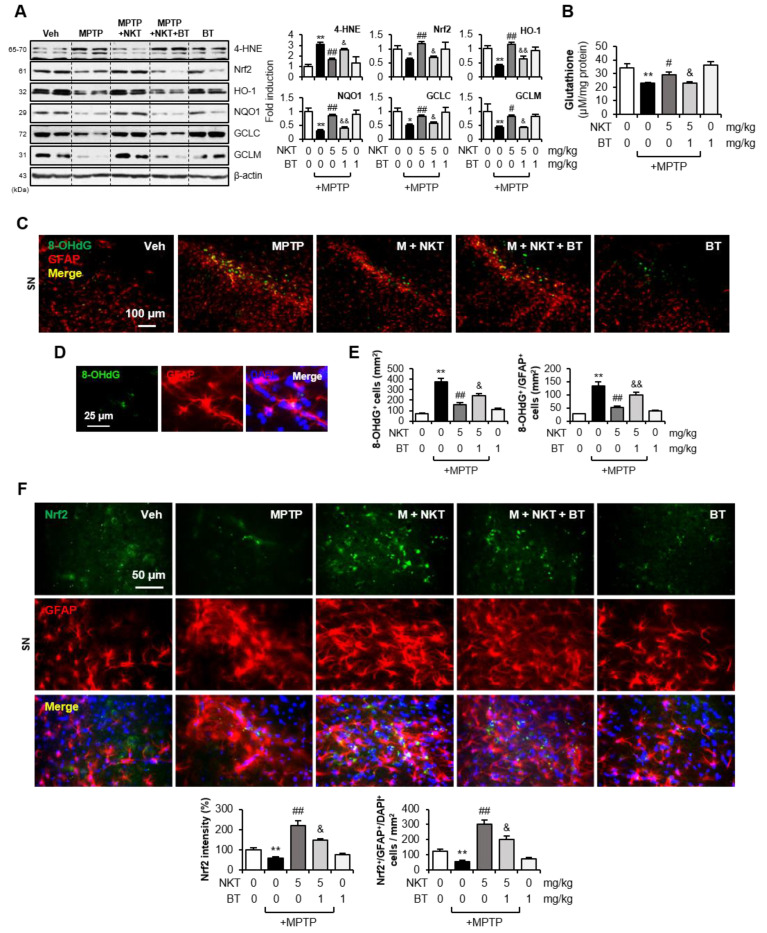
Brusatol reversed the effect of NKT on oxidative stress and astroglial Nrf2/antioxidant enzyme expression in MPTP-treated mice. (**A**) Effects of BT and NKT on the protein levels of 4-HNE, Nrf2, HO-1, NQO1, GCLC, GCLM in the SN of MPTP mice (each group n = 7–8). (**B**) Effects of BT and NKT on GSH level in the SN of MPTP-injected mice. (**C**–**E**) IF staining data showing 8-OHdG and GFAP expression in the SN of MPTP mice (n = 5–6 per group). Representative images (**C**) and high magnitude images (**D**) are provided and quantification of 8-OHdG expression and 8-OHdG^+^/GFAP^+^ cells is shown (**E**). (**F**) IF staining data showing Nrf2 and GFAP expression in the SN (n = 5–6). The upper panel has representative images and the quantification of Nrf2 expression and Nrf2^+^/GFAP^+^/DAPI^+^ cells is shown in the bottom panel. The data are presented as the mean ± SEM. * *p* < 0.05 vs. control group; ** *p* < 0.01 vs. control group; ^#^
*p* < 0.05 vs. MPTP-treated group; ^##^
*p* < 0.01 vs. MPTP-treated group; ^&^
*p* < 0.05 vs. MPTP + NKT-treated group; ^&&^
*p* < 0.01 vs. MPTP + NKT-treated group.

**Figure 8 antioxidants-12-01999-f008:**
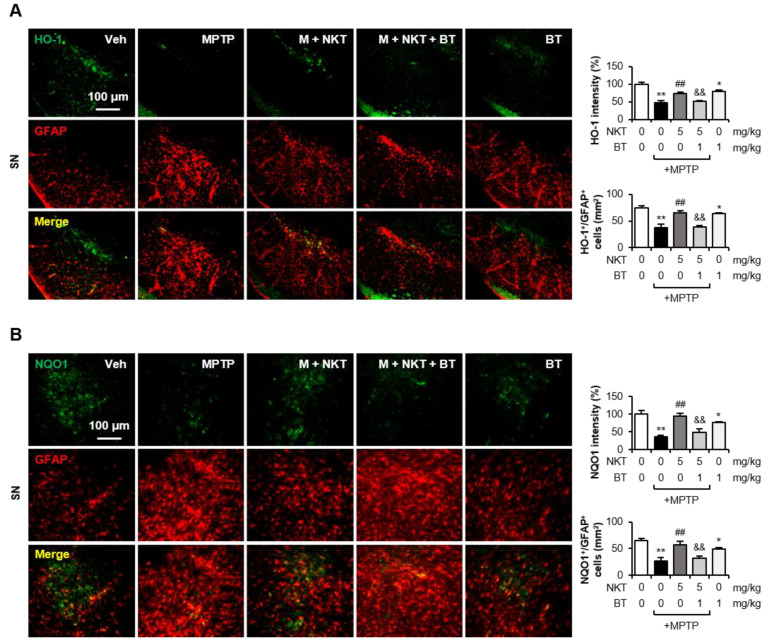
Brusatol reversed the effect of NKT on astroglial HO-1 and NQO1 expression in MPTP-treated mice. (**A**) IF staining data showing HO-1 and GFAP expression in the SN (n = 5–6, three sections per brain). The upper panel has representative images, and quantification of HO-1 expression and HO-1^+^/GFAP^+^ cells is shown in the right panel. (**B**) IF staining data showing NQO1 and GFAP expression in the SN of MPTP mice (n = 5–6 per group, three sections per brain). The left panel has representative images, and quantification of NQO1 expression and NQO1^+^/GFAP^+^ cells is shown in the right panel. Data are presented as the mean ± SEM. * *p* < 0.05 vs. control group; ** *p* < 0.01 vs. control group; ^##^
*p* < 0.01 vs. MPTP-treated group; ^&&^
*p* < 0.01 vs. MPTP + NKT-treated group.

**Figure 9 antioxidants-12-01999-f009:**
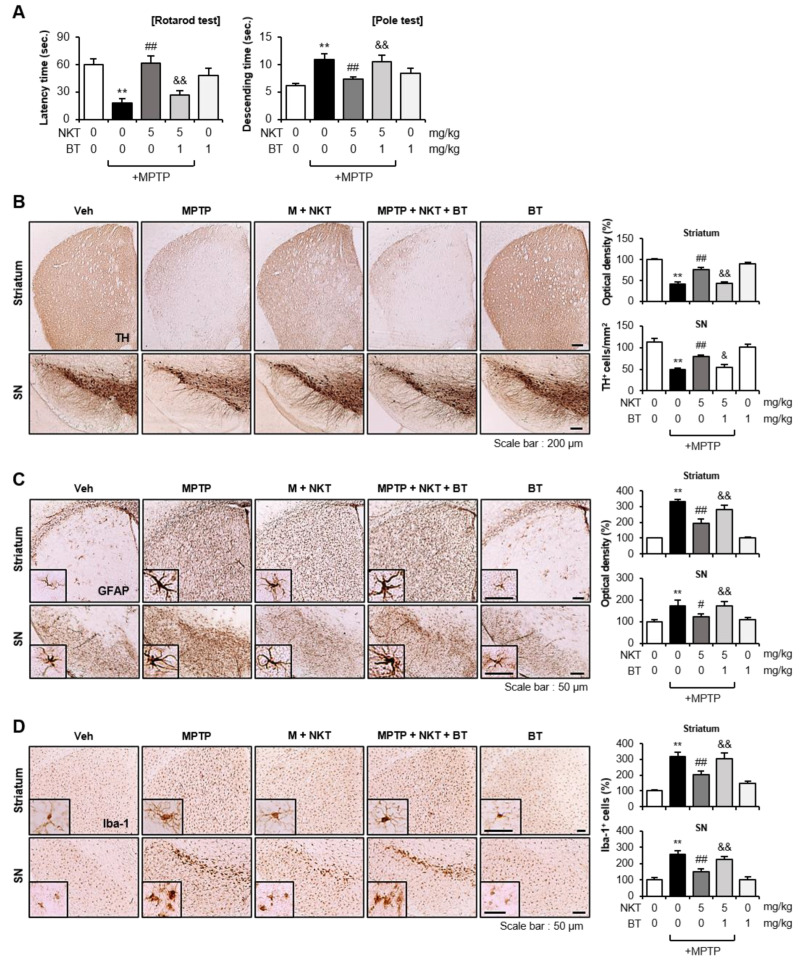
Brusatol reversed the effects of NKT on locomotor activities, dopaminergic neuronal cell death, and astrocyte/microglial activation in MPTP-treated mice. (**A**) Rotarod and pole tests were carried out two and six days following MPTP injection, respectively (each group n = 10–12). (**B**) IHC staining for TH in the striatum and SN (n = 5–6 per group). The number of TH^+^ cells in the SN and the optical density of TH^+^ fibers in the striatum were measured quantitatively (right panel). (**C**) IHC staining for GFAP in the striatum and SN (n = 5–6 per group). The optical density of GFAP^+^ cells was measured for quantitative analysis (right panel). (**D**) IHC staining for Iba-1 in the striatum and SN (n = 5–6 per group). The number of Iba-1^+^ cells was counted for quantitative analysis (right panel). The data are presented as the mean ± SEM. ** *p* < 0.01 vs. control group; ^#^
*p* < 0.05 vs. MPTP-treated group; ^##^
*p* < 0.01 vs. MPTP-treated group; ^&^
*p* < 0.05 vs. MPTP + NKT-treated group; ^&&^
*p* < 0.01 vs. MPTP + NKT-treated group.

**Figure 10 antioxidants-12-01999-f010:**
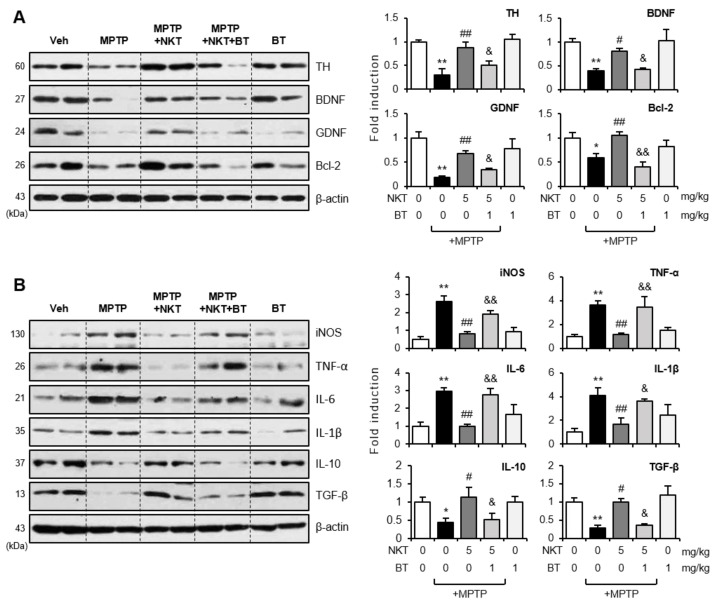
Brusatol reversed the effects of NKT on the expressions of neurotrophic factors and pro- and anti-inflammatory molecules in the brains of MPTP-treated mice. (**A**) Western blot analysis showing the effect of BT on NKT-mediated upregulation of neurotrophic factors in the SN of MPTP mice (each group n = 7–8). (**B**) Effects of BT on NKT-mediated down- or upregulation of pro- and anti-inflammatory cytokines, respectively, in the SN of MPTP mice (each group n = 7–8). The left panel shows representative blots, and the right panel shows quantification data. The data are presented as the mean ± SEM. * *p* < 0.05 vs. control group; ** *p* < 0.01 vs. control group; ^#^
*p* < 0.05 vs. MPTP-treated group; ^##^
*p* < 0.01 vs. MPTP-treated group; ^&^
*p* < 0.05 vs. MPTP + NKT-treated group; ^&&^
*p* < 0.01 vs. MPTP + NKT-treated group.

**Figure 11 antioxidants-12-01999-f011:**
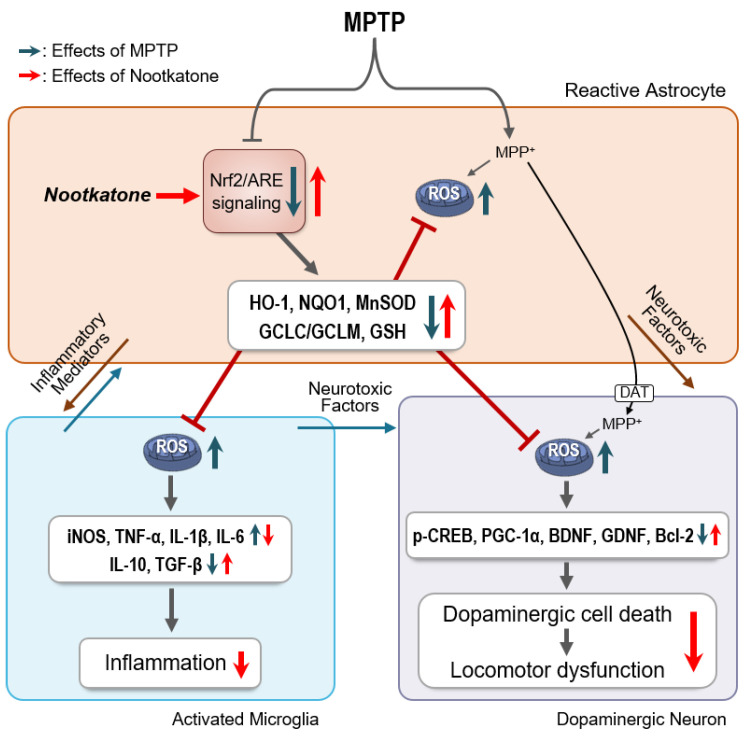
Proposed mechanism underlying the effects of NKT in an MPTP-induced PD mouse model. MPTP treatment induces dopaminergic neuronal loss and the activation of astrocytes and microglia. NKT increases Nrf2/ARE signaling and downstream antioxidant molecules, such as HO-1, NQO1, MnSOD, and GSH, in MPTP-treated astrocytes. The antioxidant molecules work together to decrease ROS production in astrocytes, microglia, and neurons, thereby inhibiting neuroinflammation and exerting neuroprotective effects.

**Table 1 antioxidants-12-01999-t001:** Primer sequences used for PCR.

Gene	Primer Sequence (5′-3′)	Size	Accession No.
*HO-1*	F: TGGCGAAGAAACTCTGTCTG R: CAACATTGAGCTGTTTGAGGA	209 bp	NM_012580
*NQO1*	F: ATCACCAGGTCTGCAGCTTC R: GCCATGAAGGAGGCTGCTGT	210 bp	NM_017000
*MnSOD*	F: GGCCAAGGGAGATGTTACAA R: GAACCTTGGACTCCCACAGA	216 bp	NM_017051
*GCLC*	F: GATGCCAACGAGTCTGACCA R: TGTAAGACGGCATCTCGCTC	470 bp	NM_012815
*GCLM*	F: AGTGGGCACAGGTAAAACCC R: CGATGACCGAGTACCTCAGC	371 bp	NM_017305
*GAPDH*	F: ACAGTCTTCTGAGTGGCAGTCA R: GTGCTGAGTATGTCGTGGAGTC	292 bp	NM_017008

## Data Availability

The datasets generated for this study are available on request to the corresponding authors.

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
