# Peer review of "Astrocytic Nrf2 Mediates the Neuroprotective and Anti-Inflammatory Effects of Nootkatone in an MPTP-Induced Parkinson’s Disease Mouse Model"

_antioxidants, 2023, doi:10.3390/antiox12111999_

Round 1

Reviewer 1 Report

Comments and Suggestions for Authors

In this paper the authors prove the protective effect of Nootkatone (NKT) against MPTP-induced Parkinson’s Disease model. They use both animals and primary astrocyte cells to carry out experiments. They examine many aspects of the question with well-planned and correctly executed elegant experiments. Their methods are modern, well-described, the consequences of the results are acceptable.

I argue with some terms used in the text: BT reverses some effect of NKT or NKT restored some activities which were lost or reduced by MPTP-treatment. To my opinion in both cases inhibition occurred, because BT was administered before NLT, and NKT before MPTP. One of my questions is related to this: is it known whether NKT have any helpful effect after MPTP-treatment?

My other questions: Was there any experiment to examine the relationship of astrocytes and microglia in the effect of NKT? Are primary astrocytes the best choice to examine the features of a disease which usually develops in elderly?  

Reviewer 2 Report

Comments and Suggestions for Authors

In this research article, the authors examined the neuroprotective effects of nootkatone, a natural compound with antioxidant properties, in one of the models of Parkinson’s disease in mice. This paper represents an intensive study, including experimental works in animals and primary cultures, where numerous parameters were tested to provide solid evidence and confirm the hypothesis. The authors describe a mechanism by which astrocytes contribute to the pathophysiology of Parkinson’s disease and a potential route to ameliorate the neuropathology via activating the expression of nuclear factor Nrf2 linked to activation of antioxidant enzymes. I support this paper for publication, pending that the authors include the necessary controls in the results section.

 1.     What was the reason for choosing the treatment scheme – drug administration for three days (Fig. 1A)? Was the drug tested at different concentrations – 2 or 5 mg/kg? What was the rationale for such a concentration chosen? 

2.     In Figure 1B, the MPTP group with no NKT treatment must be added. The statistical difference should be tested between MPTP-treated animals with NKT vs MPTP-treated animals without NKT for each time point, as shown.

3. The same applies to all graphs in Figure 1D.

4.     In Figure 2A, the analysis of the % of GFAP-positive cells (same as in Fig. 2B) would be more suitable than optical density. Astrocytes have small soma size but extended astrocytic clouds that might distort counting the signal density.

5.     Why did the authors not make primary astrocytes isolated from NKT-treated mice for the data shown in Figure 5? This should be the ideal design of the experiment performed (line 295).

6.     The Introduction needs more information about the already known role of Nrf2 in neurodegeneration. Please extend the text accordingly.

7.     Abbreviations should be used after the full name is given first (lines 71, 121, others).

8.     The limitations of the MOTO model of Parkinson’s disease should be discussed or at least mentioned.

Comments on the Quality of English Language

The text requires polishing in the academic style. There are some grammar inconsistencies and errors in English. 

Round 2

Reviewer 2 Report

Comments and Suggestions for Authors

The authors should include their explanation about the reason for the treatment scheme and that 10-50 mg/kg of NKT induced significant mortality in MPTP-treated mice, as appeared in their response. 

It is important to acknowledge and discuss the limitations of data from astrocytic cultures. This ensures that the data is not misinterpreted.

Comments on the Quality of English Language

Moderate edit. 
